# ODEdit: Blind Face Restoration through Ordinary Differential Equations

## Abstract

We introduce ODEdit, an unsupervised blind face restoration method. ODEdit operates without necessitating any assumptions about the nature of the degradation affecting the images and still surpasses current approaches in versatility. It is characterized by its utilization of the generative prior encapsulated within a pre-trained diffusion model, obviating the necessity for any additional fine-tuning or any handcrafted loss function. We leverage Ordinary Differential Equations for image inversion and implement a principled enhancing approach based on score-based updates to augment the realism of the reconstructed images. Empirical evaluations on face restoration reveal the robustness and adaptability of our methodology against a varied spectrum of corruption and noise scenarios. We further show how our approach synergies with other latent-based methods to outperform the state-of-the-art Blind Face Restoration methods in our experiments.

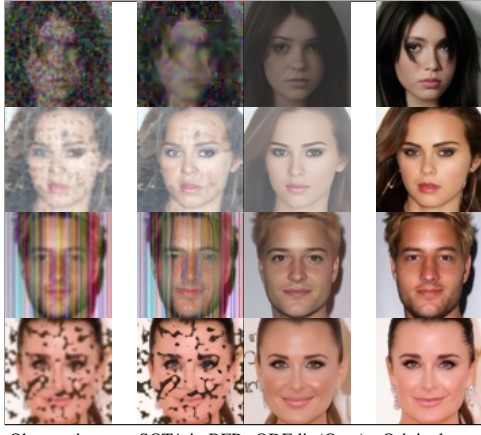

Observations     SOTA in BFR   ODEdit (Ours)   Originals

Figure 1: Ordinary Differential Equation Editing (ODEdit) is a unified Blind Face Restoration method based on ordinary differential equations. ODEdit allows to restore even severely damaged observation with a high degree of fidelity and realism.

## 1 Introduction

Blind Face Restoration consists in restoring a face image that suffer from an unknown degradation. Similarly to the more general image restoration problem, it is an inherently ill-posed problem due to the loss of information resulting from the unknown degradation. To enhance restoration quality, it is therefore necessary to incorporate auxiliary information. And the choice of the information, also called prior, to incorporate is key in the design of restoration methods. Such choice partially determine the realism of the reconstruction and its fidelity to the original observation.

Traditional image restoration techniques (Kawar et al., 2022; Choi et al., 2021; Saharia et al., 2021; 2022; Whang et al., 2022; Dhariwal & Nichol, 2021; Kadkhodaie & Simoncelli, 2020; Chung et al., 2022; Kawar et al., 2021a;b; Song et al., 2021) have been using the knowledge of the degradation

process itself as auxiliary information. They either leverage the degradation operator in pre-defined scenario such as Gaussian noise, blur or low-resolution, or, they leverage pairs of degraded and clean images to learn a mapping in a supervised fashion. While excelling in restoring specific degradation, those methods are limited in their applicability and do not transfer to real-world scenario where images exhibit various forms of unknown degradation. Hence the raise of interest for Blind Image Restoration (BIR) and Blind Face Restoration (BFR) methods that seek to extend these frontiers and whose potential applications would range from revitalizing aged photographs to providing off-the-shelf robustness for downstream visual tasks. Previous approaches in BFR (Chen et al., 2018; 2021; Yu et al., 2018) incorporated face-specific priors such as facial heatmaps or landmarks. However, such priors degrade rapidly as the degree of degradation increases, and fail to lead to versatile and robust methods. Recent approaches (Chan et al., 2021; Zhou et al., 2022; Yang et al., 2021; Pan et al., 2021; Menon et al., 2020) proposed to use instead generative prior, namely the prior encapsulated in a pr-trained generative model such as GAN (Goodfellow et al., 2014). Those methods typically rely on GAN-inversion and perform optimization in the latent space of a GAN. However, they become very unstable as the degradation increases tend to produce realistic but unfaithful reconstruction. In (Wang et al., 2021a), they proposed to overcome those limitations by combining facial priors with generative priors to improve fidelity and achieve state-of-the-art results in BFR. However, as we show in Fig. 3, they still fail to generate realistic and faithful images when the observation is too degraded. In the meantime, (Meng et al., 2021) proposed to replace GANs by diffusion models (Ho et al., 2020; Song et al., 2020c) as generative priors. They inject random noise into the degraded observation to partially invert it, in order to subsequently revert the process using stochastic differential equation (SDE). As they inject more noise, they ensure a higher degree of realism but at the expense of fidelity. We found that in the case of BFR, as the degree of degradation increases, the method requires too high a degree of noise injection to work, forcing to chose between realism or fidelity.

As of today, BFR still faces two pivotal challenges: achieving faithful and realistic reconstruction given any degraded observation.

We introduce ODEdit, an unsupervised BFR method that generates plausible and realistic face image reconstructions from a single degraded image without any prior information on the type of degradation and even in conditions of severe degradation. ODEdit relies on the generative prior of a pre-trained diffusion model.
Similarly to GAN-inversion methods (Tov et al., 2021; Abdal et al., 2020; 2019; Zhu et al., 2020; Menon et al., 2020; Pan et al., 2021), the observation is first inverted to the latent space before performing optimization. While GAN-inversion relies on a auxiliary trained encoder, diffusion model inversion relies on differential equations. Consequently, as opposed to GAN-inversion methods, ODEdit requires no extra training and is more robust to out-of-distribution observations. Furthermore, ODEdit, builds upon SDEdit (Meng et al., 2021), by replacing their inversion method. While they simulate inversion through noise injection, we use probability-flow Ordinary Differential Equation (Song et al., 2020c), which maintains a one-to-one mapping with the latent space. Consequently, ODEdit ensures a higher degree of fidelity between our reconstruction and the observation compared to SDEdit. As a result, ODEdit surpasses current approaches in term of fidelity and realism when facing a broad spectrum of degradation. To the best of our knowledge, ODEdit is the first unsupervised approach to use ODE-based inversion in Diffusion Models for Image restoration. Finally, building upon the idea of combining priors introduced in (Wang et al., 2021a; Lin et al., 2023), we propose to combine our approach with a powerful latent-based generative model, by inputting to it the restored outputs of ODEdit. Such hybrid approach, that we denote ODEdit+, proves beneficial in reconstructing fine-grained details and outperforms all recent BFR methods on CelebA (Liu et al., 2015).

## 2 RELATED WORKS

**Inverse Problem** In the inverse problem setups, models are designed to leverage the knowledge of linear degradation operators. When amalgamated with potent generative models like diffusion models, these methods have exhibited unparalleled results, setting new benchmarks (Choi et al., 2021; Saharia et al., 2022; Liang et al., 2021; Kawar et al., 2022). However, the prerequisite of exact knowledge of the degradation operator, while yielding exceptional results in applications like

deblurring or super-resolution, is often unrealistic in real-world scenarios and are therefore not directly applicable to our scenario.

**Conditional Generative Models**    A parallel approach involves conditioning a generative model on a degraded image. While in most approaches (Mirza & Osindero, 2014; Isola et al., 2017; Batzolis et al., 2021), this requires pairs of degraded and natural images, it may also rely on the design of loss functions and guidance functions as shown in (Song et al., 2020c). Conditional Generative Adversarial Networks, as explored in (Isola et al., 2017), exemplify this approach, where generative models are trained to regenerate the original sample when conditioned on its version in another domain. However in the case of unknown and diverse degradation, such methods while inspiring are not applicable and are not really comparable.

**Unsupervised Bridge Problem**    In scenarios where two distinct datasets of clean and degraded data are available without direct paired data, methodologies relying on principles like cycle consistency and realism have been developed, as evidenced by the works of (Zhu et al., 2017) using GANs (Goodfellow et al., 2014) and (Su et al., 2023) using Diffusion Models (Ho et al., 2020),(Sohl-Dickstein et al., 2015). A direct application of such methods to our scenario is not feasible, as we target unknown degradation in a zero-shot fashion.

**Blind Image Restoration**    Akin to our proposed methods are blind image restoration approaches. Blind Image Restoration methods do not restrict themselves to specific degradations. One recent approach has been to transpose the problem into a latent space where correction based on the prior distribution are being performed. The works by (Abdal et al., 2019; 2020; Chan et al., 2021; Zhu et al., 2020) have explored various aspects of this approach, utilizing GAN inversion, or VAE/VQVAE encoding (Kingma & Welling, 2013; Oord et al., 2017), and have achieved significant advancements, particularly in scenarios involving light but diverse degradations. A prominent work is CodeFormer introduced by (Zhou et al., 2022), which leverages the potential of transformer in the latent space of a pre-trained VQ-VAE. However,the most dominant methods (Wang et al., 2021a; Gu et al., 2022; Zhou et al., 2022) rely on the combination of many blocks trained separately, such as face detector or background enhancer (Wang et al., 2021b), making it even harder to ensure resilience to diverse degradations.

A novel approach by (Meng et al., 2021) has surfaced, focusing on the stochastic exploration of the neighborhood of a given input to yield outputs that are both realistic and faithful. This method, tested for robustness by (Gao et al., 2022), bears similarities to the proposed gradient updates within the latent space of GAN models but is grounded in more solid theoretical foundations.The method is not orthogonal to latent-based approaches, as SDEdit has been applied within the latent space of Stable Diffusion (Rombach et al., 2022). We follow the success of SDEdit and propose a method alleviating some of their shortcomings.

Finally, a inspiring recent work by (Lin et al., 2023) has introduced a hybrid approach, integrating a pixel-based restoration module with a latent-based diffusion model, offering more refined and detailed image reconstructions. Our method falls within this category as it shows best performance when being used as a first restoration stage before powerful latent-based ones.

## 3    METHOD

### 3.1    PRELIMINARY

Diffusion models (Sohl-Dickstein et al., 2015; Ho et al., 2020) are a class of generative models that simulate a diffusion process to transform data, typically images, from one state to another. They generate data by reversing the forward diffusion process that transforms original data points into noise.

**Denoising Diffusion Probabilistic Models**    We denote $\mathbf{x}_0$ the data from the data distribution, in our case natural images, and $\mathbf{x}_1, ..., \mathbf{x}_T$ the latent variables. The forward process is in DDPM (Ho et al., 2020) then defined by:

$$\mathbf{x}_{t+1} = \alpha_t \cdot \mathbf{x}_t + (1 - \alpha_t) \cdot \epsilon, \text{with } \epsilon \sim \mathcal{N}(0, \mathbf{I}). \tag{1}$$

where $\alpha_t$ is a schedule predefined as an hyperparameter.

The diffusion model $\epsilon_\theta$ is then trained to minimize

$$\mathbb{E}_{\mathbf{x}_t, t} \|\epsilon_\theta(\mathbf{x}_t, t) - \epsilon\|. \tag{2}$$

**Score-based Generative Models** In the case of Score-based Generative Models (Song & Ermon, 2019; Song et al., 2020c; Song & Ermon, 2020), the model $s_\theta$ learns to approximate the score function, $\nabla_\mathbf{x} \log p(\mathbf{x})$, by minimizing

$$\mathbb{E}_{p(\mathbf{x})} \|s_\theta(\mathbf{x}) - \nabla_\mathbf{x} \log p(\mathbf{x})\|. \tag{3}$$

In most scenarios, the ground truth score function is unavailable but fortunately there exists a class of methods denoted Score Matching (Hyvärinen, 2005; Vincent, 2011; Song et al., 2020b) to minimize the above divergence without access to the ground truth score values. The most commonly used approach is denoising score matching (Vincent, 2011). In this framework, the data $\mathbf{x}$ is first perturbed by a Gaussian noise. If we note the noised distribution $\tilde{\mathbf{x}}$ then: $q_\sigma(\tilde{\mathbf{x}}|\mathbf{x}) = \mathcal{N}(\tilde{\mathbf{x}}, \mathbf{x}, \sigma^2 \mathbf{I})$ then $\nabla_{\tilde{\mathbf{x}}} \log q_\sigma(\tilde{\mathbf{x}}|\mathbf{x}) = -\frac{\tilde{\mathbf{x}}-\mathbf{x}}{\sigma^2}$ and the denoising score matching objective becomes:

$$\mathbb{E}_{p(\mathbf{x})} \mathbb{E}_{\tilde{\mathbf{x}} \sim \mathcal{N}(\mathbf{x}, \sigma^2 \mathbf{I})} \|s_\theta(\tilde{\mathbf{x}}, \sigma) - \frac{x - \tilde{x}}{\sigma^2}\|. \tag{4}$$

Crucially one can sample from $p(x_t)$ while using only the score function through Langevin dynamics (Langevin, 1908) sampling, by repeating the following update step:

$$x_{t+1} = x_t + \epsilon \cdot s_\theta(x_t, t) + \sqrt{2\epsilon} \cdot \eta \text{ , with } \eta \sim \mathcal{N}(0, \sigma^2). \tag{5}$$

**Differential Equation Framework** Diffusion models can also be expressed in continuous time using a forward Stochastic Differential Equation (SDE) and its reverse counterpart (Anderson, 1982):

$$d\mathbf{x} = \mathbf{f}(\mathbf{x}, t)dt + g(t)d\mathbf{w}, \ d\mathbf{x} = [\mathbf{f}(\mathbf{x}, t) - g(t)^2 \nabla_\mathbf{x} \log p_t(\mathbf{x})]dt + g(t)d\mathbf{w}. \tag{6}$$

Where $\mathbf{w}$ corresponds to the standard Wiener process, $g(t)$ is the diffusion coefficient, $\mathbf{f}(\mathbf{x}, t)$ is the drift coefficient and $\nabla_\mathbf{x} \log p_t(\mathbf{x})$ is the score function mentioned above.
In the case of SDEdit, they leverage this reverse SDE to generate sample from a coarse image. However, (Song et al., 2020c) shows that there exists a unique ordinary differential equation (ODE) for which the distribution of the latent variable at any time t equals the distribution generated by the SDE. They called this ODE, the probability-flow ODE whose equation is the following:

$$d\mathbf{x} = [\mathbf{f}(\mathbf{x}, t) - \frac{1}{2}g(t)^2 \nabla_\mathbf{x} \log p_t(\mathbf{x})]dt. \tag{7}$$

Using ODE instead of SDE not only makes the solver more accurate and invertible. Using the ODE to invert and reconstruct precisely the original input is the cornerstone of our method.

### 3.2 ODEDIT: EDITING VIA ORDINARY DIFFERENTIAL EQUATIONS

Our approach, ODEdit, integrates diverse facets of diffusion modeling to formulate a theoretically substantiated method for blind restoration. It uses the relationship between Stochastic Differential Equations (SDE) and their corresponding Ordinary Differential Equations (ODE), leverages the equivalence between score-based generative models and denoising diffusion probabilistic models, and exploits the optimal-transport properties of the inversion to implicitly provide auxiliary guidance for restoration.

**From Stochastic Hijack to Ordinary Inversion** While our method draws inspiration from SDEdit (Meng et al., 2021), it significantly differs from their 'hijacking' of the diffusion process which involves injecting noise into the observation. In their paper, the authors propose inverting the diffusion process by directly injecting Gaussian noise into the input image. They then use this newly created noisy image as a starting point to generate an image using DDPM (Ho et al., 2020) , with the number of steps determined by the amount of noise reintroduced. There exists a trade-off between fidelity and realism, the more noise being injected the more realistic the reconstructed image will

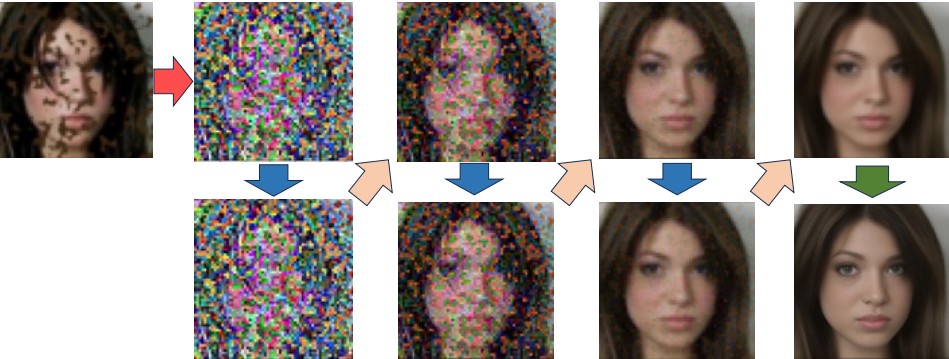

Figure 2: **ODEdit.** We first invert our degraded observation back to the latent space (red arrow). Then we alternatively enhance the latent using Langevin dynamics (blue arrow) and use the ODE solver to revert back to the following latent space (orange arrow). Finally, we use an off-the-shelf upsampler to restore our output in high-resolution (green arrow).

be, but at the expense of fidelity to the original input. There has been attempts to use SDEdit for denoising unknown corruption but it often requires extra guidance or assumptions, and the results have been limited (Gao et al., 2022). Instead, we propose the inversion of the Probability-Flow Ordinary Differential Equation (ODE) as a superior alternative. This modification yields advantages regarding the fidelity of the reconstructed image to the observed data as we circumvent the injection of extra noise. Importantly, the inversion process, along with its reverse operation, guarantees precise image reconstruction, limited only by approximation errors (Su et al., 2023). Unlike SDEdit, which necessitates escalating levels of noise to revert to preceding latent spaces, our technique allows for a full inversion to the original latent space, while maintaining the integrity of the restored image.

In our experiments, we use the particular ODE solver from DDIMs (Song et al., 2020a) given by the following equation:

$$\frac{\mathbf{x}_{t-\Delta_t}}{\sqrt{\alpha_{t-\Delta_t}}} = \frac{\mathbf{x}_t}{\sqrt{\alpha_t}} + \left( \sqrt{\frac{1-\alpha_{t-\Delta_t}}{\alpha_{t-\Delta_t}}} - \sqrt{\frac{1-\alpha_t}{\alpha_t}} \right) \epsilon_\theta(\mathbf{x}_t, t) . \tag{8}$$

However, we note that there exists other ODE solvers (Karras et al., 2022; Zhang & Chen, 2022; Lu et al., 2022) that could be used in our framework without further refinements to potentially improve results. It is worth noting that the ODE solver is used forward and backward as in the limit of infinitesimal steps it is equal. However, it leads in practise to small approximations errors.

The primary motivation for inverting the degraded image derives from the model's capability to process out-of-distribution images. A direct estimation of the Stein score on degraded images is theoretically desirable; however, it is impractical due to the inferior performance of deep learning models on such out-of-distribution data. Conversely, our model trained in the latent space, functions within a distribution that closely mirrors a multivariate Gaussian. By transposing the corrupted input back to this latent space via Ordinary Differential Equations (ODE), we can procure more precise estimates within that space prior to re-mapping. This concept underpinned the methodology of SDEdit; however, their dependence on noise injection and Stochastic Differential Equations (SDEs) precluded the possibility of inverting the entire diffusion process without losing the inherent information in the observation. ODEdit ameliorates such deficiencies, especially evident in instances of severely degraded observations.

**From Diffusion Models to Optimal Transport**  In (Chen et al., 2022), the intricate relationship between score-based generative models (SGM) and the Schrödinger Bridge Problem (SBP) (Léonard, 2013; Chen et al., 2016) is elucidated, as they demonstrate that Score-Based Generative models serve as implicit optimal transport models. As our method exclusively employs score-based updates for image restoration, it inherently associates with such an optimal transport problem in pixel-space. Our ODE solver minimizes the mass transport from our latent to the visual domain. Hence, while we explore the neighborhood, in distance pixel-wise, in our latent space, the result-

ing output is itself not far from the original observation. This association provides insights on the mechanics, and constraints of our method. Our method is a principled way to explore around the corrupted observation to identify a more probable but low optimal transport cost image. In essence, our approach endeavors to minimize alterations pixel-wise while enhancing image quality.

As a result ODEdit frames the restoration task into an optimization problem, solving it using Langevin dynamics. Our methodology embodies a common principle in restoration, seeking natural images close to our observation, with a concept of proximity based on pixel-to-pixel distance.

**From DDPM to Stein-Score**   A key factor of our approach lies in its strategic utilization of the relationship between score-based diffusion models and DDPM. Specifically, we employ only one model trained via the DDPM procedure for multiple purposes: to invert the corrupted input using Denoising Diffusion Implicit Model(DDIM) as our probability-flow inversion technique, to estimate the Stein scores at intermediate states, and to execute deterministic sampling through DDIM. There exists a direct correspondence between DDPM, $\epsilon_\theta$, and Noise Conditional Score Network, $s_\theta$, such that:

$$s_\theta(\mathbf{x}_t, t) = -\frac{\epsilon_\theta(\mathbf{x}_t, t)}{\sigma_t}. \tag{9}$$

**Latent score-based updates**   Building upon that correspondence, we propose to perform gradient-update in our intermediate latent spaces utilizing Langevin dynamics as in equation equation 5. Thus, we first perform inversion using DDIM as the ODE solver to obtain a latent code. The likelihood of this latent code is linked to the likelihood of the image through equation 10 (Feng et al., 2023).

$$\log p_0(\mathbf{x}_0) = \log p_T(\mathbf{x}_T) + \int_0^T \nabla \cdot \mathbf{f}(\mathbf{x}, t) - \frac{1}{2}g(t)^2 s_\theta(\mathbf{x}_t, t) dt. \tag{10}$$

As this equation remains true for any T, we propose improving the likelihood of the generated image by performing score-based updates in the intermediary latent spaces, extending all the way back to the visual domain, starting from the latent code obtained with DDIM. Thus, we alternate between score-based updates and DDIM steps to generate the restored image. While it might seem feasible to update only our latent code and then use DDIM to generate an image, the alternation has proven more effective in producing realistic images.

The whole method is described in Alg. 1. These updates bear a clear resemblance to those employed by traditional score-based generative models, albeit with several noteworthy distinctions. Firstly, we initiate the process with the inverted input image and not a random vector. Secondly, we do not use annealed Langevin dynamics to converge to the natural image distribution. Instead we use minor gradient step to refine our latent, relying on the ODE solver to map it back to the visual domain. These changes allow ODEdit to converge to the most likely image in the neighborhood of the inverted observation, ensuring an better trade-off in fidelity and realism. Similarly to SDEdit and as opposed to other approaches, ODEdit may be adapted to favor realism over fidelity and vice-versa by adjusting the number of updates performed.

---

**Algorithm 1** ODEdit algorithm

---

**Require:** $K$ *(Langevin iterations)*, $\{\sigma_i^2\}_{i=1}^N$ *(Variance Schedule)*, $N$ *(ODE-solver steps)*, $\epsilon$ *(stepsize)* , $x_0^g$ *(Observation)* , $T$ *(Temperature)*.
    $x_{N,0} = ODE\_SOLVER_{forward}(x_0^g)$
    **for** $t = N$ **to** $1$ **do**
        $\alpha_t \leftarrow \epsilon \cdot \frac{\sigma_t}{\sigma_1}$
        **for** $k = 0$ **to** $K - 1$ **do**
            $x_{t,k+1} = x_{t,k} - \alpha_t \cdot \frac{\epsilon_\theta(x_{t,k},t)}{\sigma_t} + \sqrt{2\alpha_t T} \cdot \eta$ , where $\eta \sim \mathcal{N}(0, \mathbf{I})$
        **end for**
        $x_{t-1} = ODE\_SOLVER_{backward}(x_{t,K})$
    **end for**

---

**Latent based refinement and upsampling**   Similarly as the method proposed in (Lin et al., 2023), we advocate for a hybrid approach . Initially, our method performs restoration at a low resolution

capitalizing on pixel-wise information, thereby partially mitigating the challenges associated with out-of-distribution data. Subsequently, these low-resolution restored outputs may easily serve as inputs to state-of-the-art latent-based upsampler algorithms.

# 4 EXPERIMENTS

## 4.1 EXPERIMENTAL SETUP

We employ the U-Net architecture as defined in (Dhariwal & Nichol, 2021), training a 120 million-parameters model on the CELEBA dataset, which comprises about 200,000 aligned and centered faces. The model is trained and tested on low-resolution images of dimensions $64 \times 64$ pixels. During training, we For inversion processes, DDIM (Song & Ermon, 2020) is utilized with 100 steps. Enhancement is achieved using Langevin Dynamics, where $K = 100$ represents the number of Langevin steps, the step size is set to $\epsilon = 1 \times 10^{-6}$, and the temperature is set to $\frac{1}{3}$. Langevin sampling is conducted in 4 out of the 100 latent spaces at times $t = 0, 25, 50, 75$.

The degree of faithfulness of our method can be modulated by varying the number of update steps, with 100 steps for light corruption, 200 for medium corruptions, and 300 for heavy ones.
Corruptions are based on the definitions by (Hendrycks & Dietterich, 2019), allowing also for variations in degradation severity. The types of corruptions is board including compression artefacts, blur, elastic transforms, contrast changes, ... We also introduce two additional masking types to assess versatility: one masking entire vertical lines and the other masking random pixels, both with random colors. Unlike traditional masking in masked autoencoders (He et al., 2022), knowledge of which pixels are masked is not assumed, posing a more realistic recovery task. The severity of these two corruptions can be finetuned by changed the percentage of pixels modified. In our quantitative study, we use CelebA-test set that comprised about 20,000 images. We randomly applied a degradation with severity 4 ((Hendrycks & Dietterich, 2019)) to the inputs. In our qualitative evaluations, DiffBir (Lin et al., 2023) is employed as our second stage.

## 4.2 BASELINES

To benchmark our approach, we evaluate its performance against several state-of-the-art methods. Initially, we illustrate the enhancements our model offers over SDEdit (Meng et al., 2021). As their method is stochastic and depends greatly on its hyperparameters, we ran SDEdit with several sets of hyperparameters and only display the best sample. Subsequently, we draw comparisons with the recently proposed DiffBIR (Lin et al., 2023), which also advocates a two-stage approach. Further, we juxtapose our method with CodeFormer (Zhou et al., 2022), which employs VQVAE and conducts denoising exclusively in the latent space utilizing Transformer (Vaswani et al., 2017). Additionally, we compare our approach with a GAN-inversion method (Tov et al., 2021) that performs updates in the GAN latent space. Lastly, we incorporate comparisons with GFPGAN (Wang et al., 2021a), a leading GAN-based face restoration method.

We primarily present our results on low-resolution images, demonstrating that, although fine-grained details are absent, ODEdit effectively mitigates the principal degradation. Subsequently, we display the outcomes of ODEdit when upscaled and further refined using DiffBIR. In terms of restoration time, our method is comparable to SDEdit, as it invokes the model a similar number of times.

## 4.3 RESULTS

We first evaluate our results with regard to the non-degraded original image, using the following metrics: PSNR, SSIM, and LPIPS. We also study the quality of our generations using FID. Those results are visible in Table 1. We observe that our method is on par with other SOTA methods in terms of PSNR, SSIM and LPIPS, and that it significantly outperforms all of them in FID metric. As a result, our method is the only one to produce realistic restoration while keeping the same degree of fidelity as other methods. Furthermore, ODEdit is an optimization method that we have specifically constrained to remain close to the source image in terms of pixel-wise similarity. This significantly limits the method's performance in terms of pixel-based metrics like PSNR or SSIM. However, it does not notably impact its performance in terms of FID or perceptual quality (also known as LPIPS). Therefore, the results align with the intended design of the method. As expected,

SDEdit is the only method which also improves FID compared to the observation but at the expense of fidelity. We then display visual comparison results in Fig. 3 and in Appendix A.2. We show that ODEdit and SDEdit seem to be the only methods guaranteeing a fair amount of realism across all corruptions. It comes naturally as both methods can adapt their hyperparameters to the degree of degradation and trade-off faithfulness to the degraded observation with realism. However, ODEdit manages to maintain a higher degree of faithfulness with the same degree of realism.

Table 1: Quantitative comparison of Restoration Methods on random degradation

|  | LPIPS ↓ | SSIM ↑ | PSNR ↑ | FID ↓ |
|---|---|---|---|---|
| *Source* | 0.34 | 0.56 | 18.01 | 83.02 |
| *GFPGAN* (Wang et al., 2021a) | 0.25 | 0.61 | 18.41 | 116.76 |
| *DiffBIR* (Lin et al., 2023) | 0.28 | 0.61 | **18.68** | 172.64 |
| *CodeFormer* (Zhou et al., 2022) | 0.26 | **0.62** | 18.53 | 157.38 |
| *SDEdit* (Meng et al., 2021) | 0.30 | 0.52 | 16.03 | 45.85 |
| ***ODEdit - ours*** | **0.24** | 0.60 | 17.8 | **33.81** |

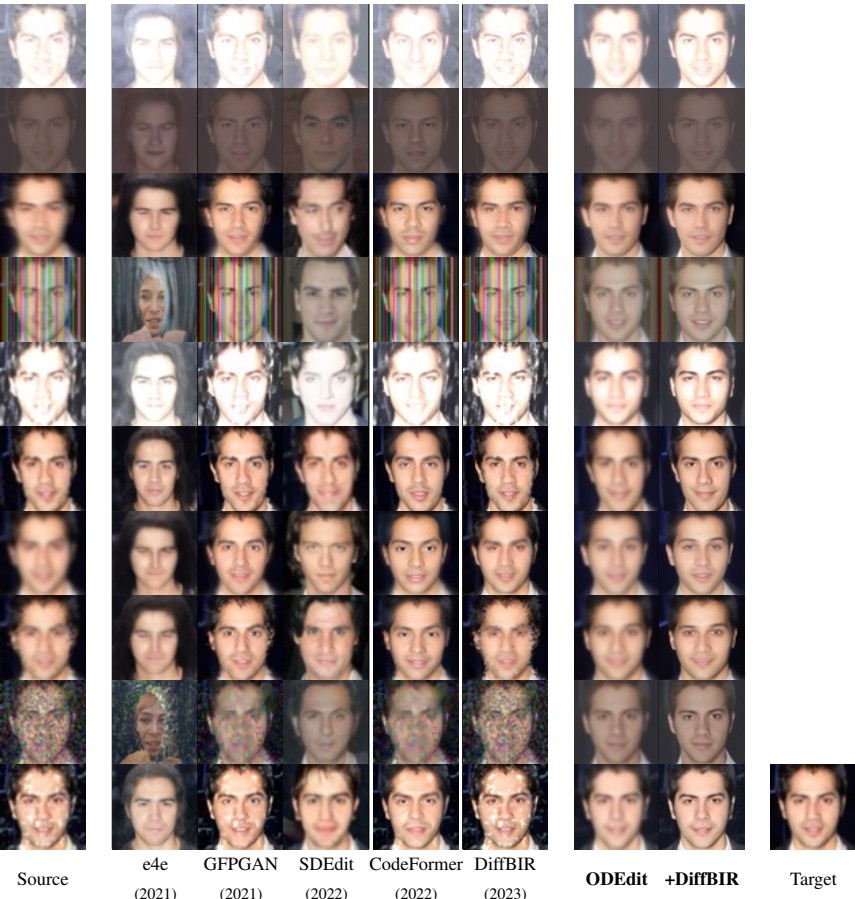

Figure 3: Visual comparison of Blind Face Restoration on various medium corruptions

## 4.4 SYNERGY STUDY

We study ODEdit's compatibility with various enhancement methods in Fig. 4. To investigate this, we subject a set of corrupted images to preliminary correction using ODEdit at low resolution. We then employ diverse enhancement methods to refine our results to high resolution. This demonstrates that, whereas other approaches may struggle to directly produce realistic images from corrupted

sources, employing ODEdit as an initial step ensures consistency and functions comparably across the three tested methods. This illustrates the quality of ODEdit's outputs, which can be seamlessly integrated into various super-resolution methods.

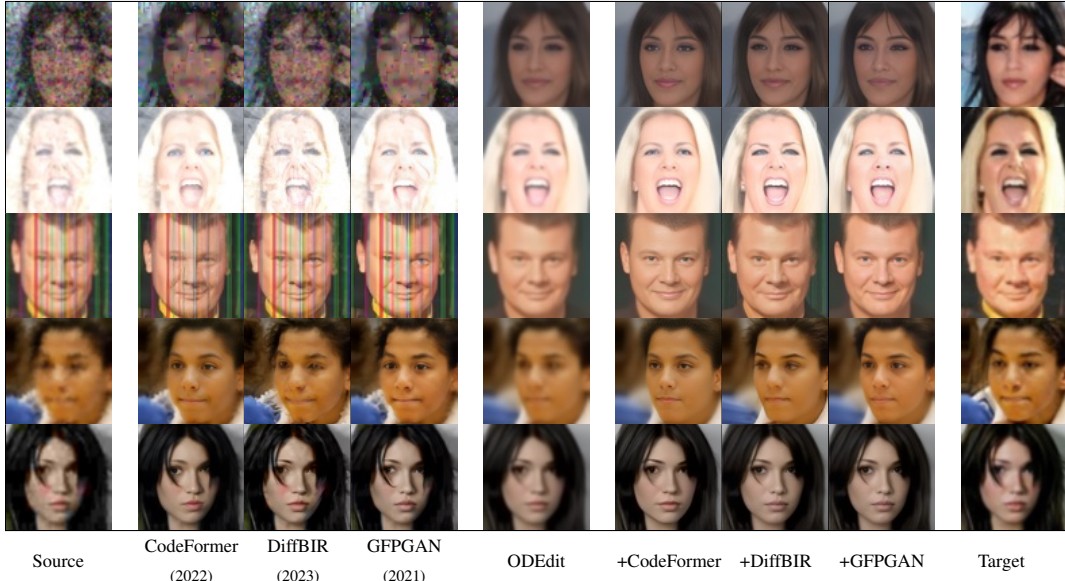

| Source | CodeFormer (2022) | DiffBIR (2023) | GFPGAN (2021) | ODEdit | +CodeFormer | +DiffBIR | +GFPGAN | Target |

Figure 4: Comparison of results using different off-the-shelf methods. The first column is the degraded observation. Columns 2 to 4 are the other methods. In column 5 we show the superiority of ODEdit in removing degradation. Columns 6 to 8 are the results are combining ODEdit and other methods.

## 5 DISCUSSION AND FUTURE WORK

One of the current limitations to our method pertains to the process of inversion. It is widely acknowledged that the current state of art in reversing diffusion processes remains unstable. Presently, we observe a degree of stability in inverting low-resolution faces hence our focus. However, when applied to more diverse domains or higher resolution images, the inversion has shown a high degree of instability leading to inaccurate reconstructions. While our study stands as a proof of concept for now, we anticipate that as advancements are made in the inversion of diffusion models, our method could extend to higher-dimensional spaces.

Another noteworthy limitation, which can be viewed as an underlying assumption, pertains to the challenge posed by color alterations. Given the pixel-centric nature of our method, it encounters difficulties when confronted with substantial alterations in color gradients. Specific examples illustrating these limitations are provided in the appendix A.3. To address this constraint, several approaches may be explored. This could first be left to other approaches altogether, as an extra-refinement step. Another interesting line of research would be to integrate other sort of metric rather than relying on a pixel-to-pixel one.

## 6 CONCLUSION

In this work, we introduce a principled way to restore degraded faces in the absence of any additional guidance that not only surpasses other methods in dealing with extreme and versatile corruptions but also shows a natural and impressive synergy when combined with other methods. As so, our work stands as a testament to the potential advancements in the field of Blind Face Restoration, offering a solution that is not only robust and versatile but also insightful. While currently limiting our study to faces, we hope our method will pave the way for future research relying on Diffusion Model inversion, similarly to how GAN inversion has once proven to be an promising field of study.

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

## A  APPENDIX

### A.1  RESULTS ON 256X256 IMAGES

#### A.1.1  COMPARISON WITH OTHER METHODS

In this section, we present promising results on 256x256 images. Assuming stability in the DDIM inversion, ODEdit performs well at higher resolutions without the need for supplementary approaches. The remaining challenge is the inherent instability of DDIM when used as an ODE solver. In this high-resolution setup, we showcase the outcomes of applying ODEdit directly, without the involvement of any additional methods.

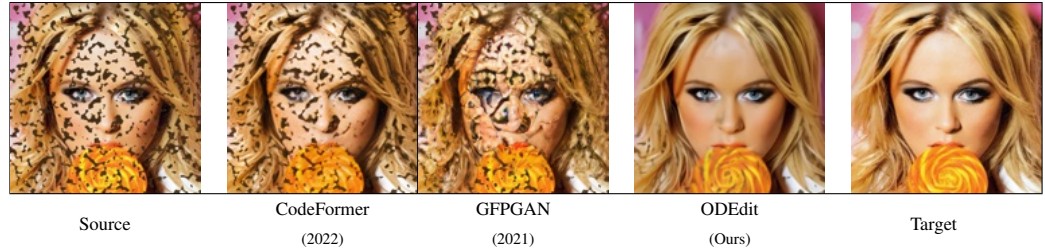

| Source | CodeFormer (2022) | GFPGAN (2021) | ODEdit (Ours) | Target |

Figure 5: Comparison of results using different methods on 256x256 face.

To demonstrate the superiority of ODEdit over SDEdit in these scenarios, we propose displaying how SDEdit functions depending on the number of updates performed, and comparing these results with those obtained by ODEdit using a varying number of updates. While both methods produce satisfactory outputs with the appropriate number of updates, ODEdit consistently maintains a higher degree of fidelity while achieving a similar level of realism.

We also demonstrate that ODEdit does not merely perform a simple average over inputs. This is done by comparing the results of using a blurring algorithm in place of ODEdit and then feeding the blurred input to other methods.

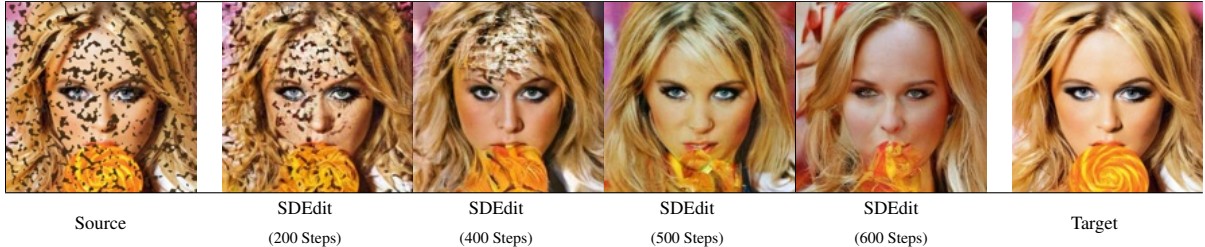

Figure 6: SDEdit on 256x256 face.

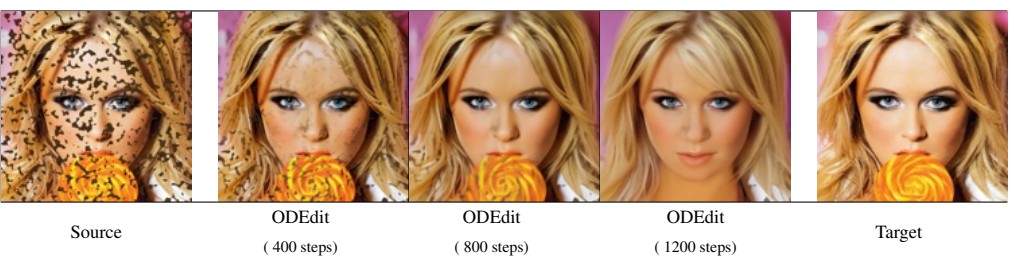

Figure 7: ODEdit on 256x256 face.

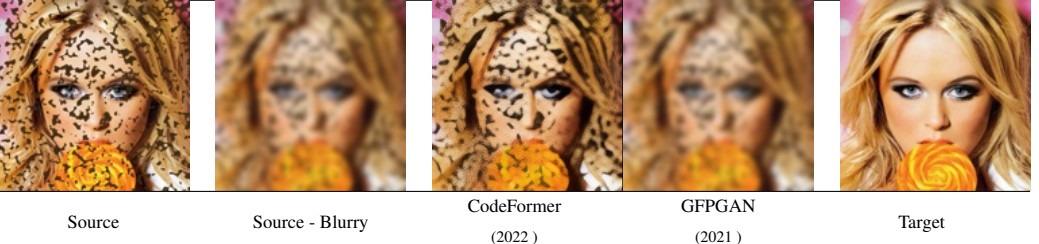

Figure 8: Performance of others methods on Blurred Inputs

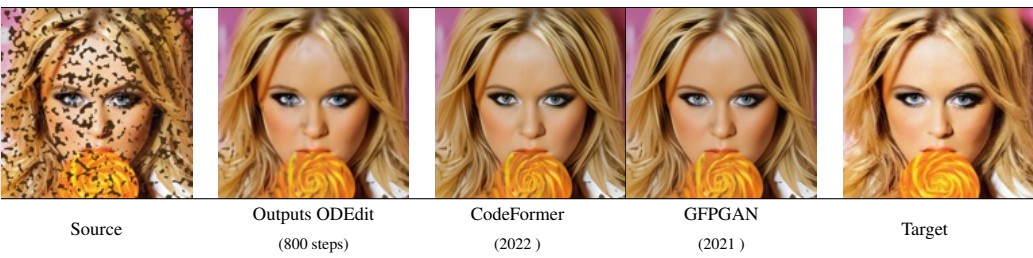

Figure 9: Performance of others methods on ODEdit Outputs

### A.1.2 Additional Results

In Fig 10 we present an example of real-world restoration. It should be noted that our model has limited training and has never encountered monochrome images. Furthermore, ODEdit, which relies on optimal transport, inherently lacks the ability to handle coloration or deal with changes in color effectively. However, we believe that despite these limitations, it demonstrates interesting results.

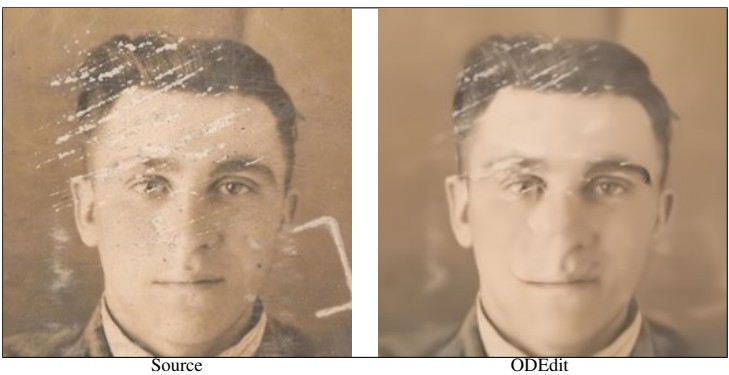

Source                                                    ODEdit

Figure 10: ODEdit on old restoration task

## A.2 Additional Results on 64x64 images

Additional results on different degrees of corruption are available in Fig. 11 and in Fig. 12.

## A.3 Limitations

Examples of failure cases are shown in Fig. 13.

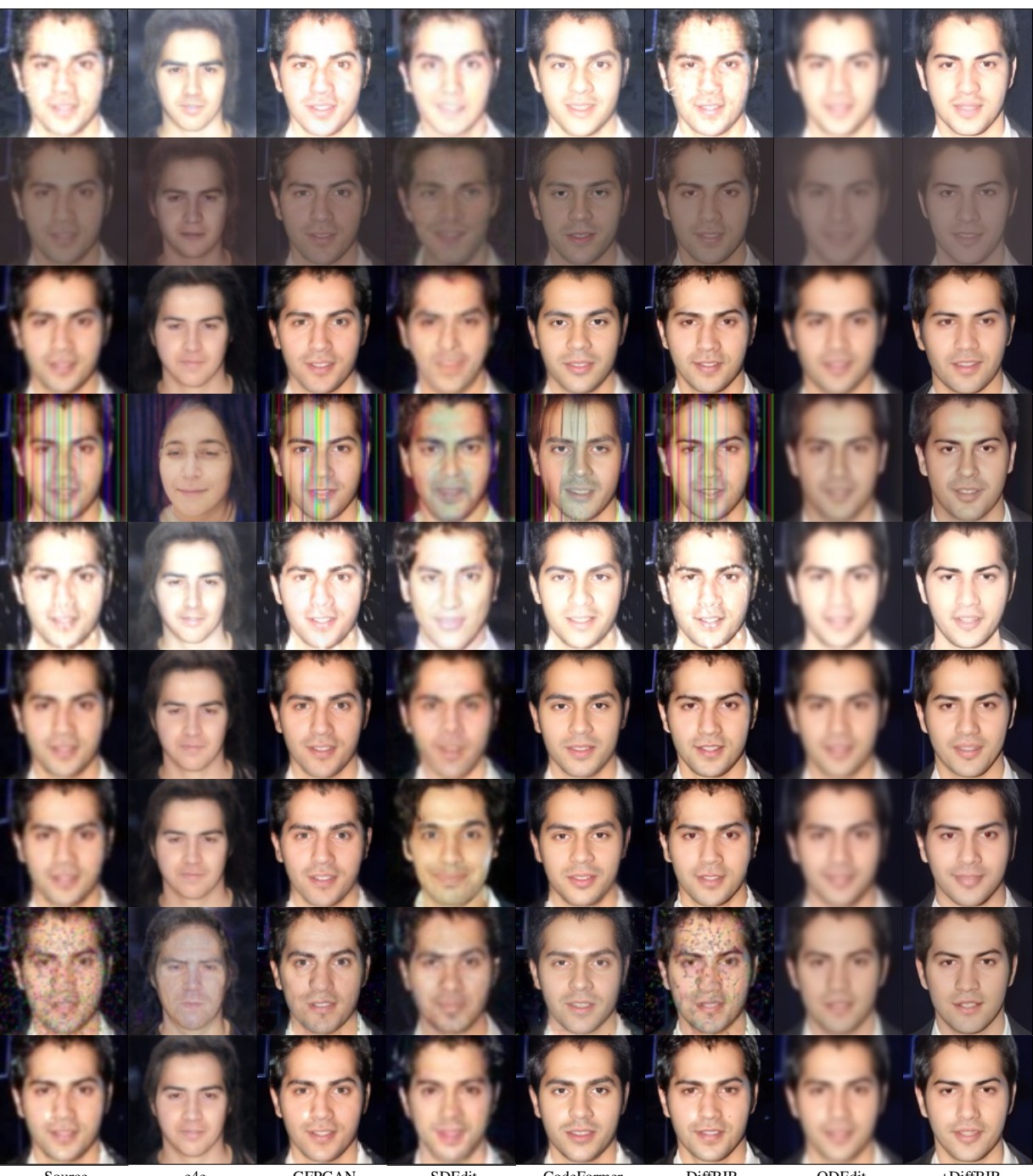

Source    e4e    GFPGAN    SDEdit    CodeFormer    DiffBIR    ODEdit    +DiffBIR

Figure 11: Visual comparison of Blind Face Restoration on light corruptions

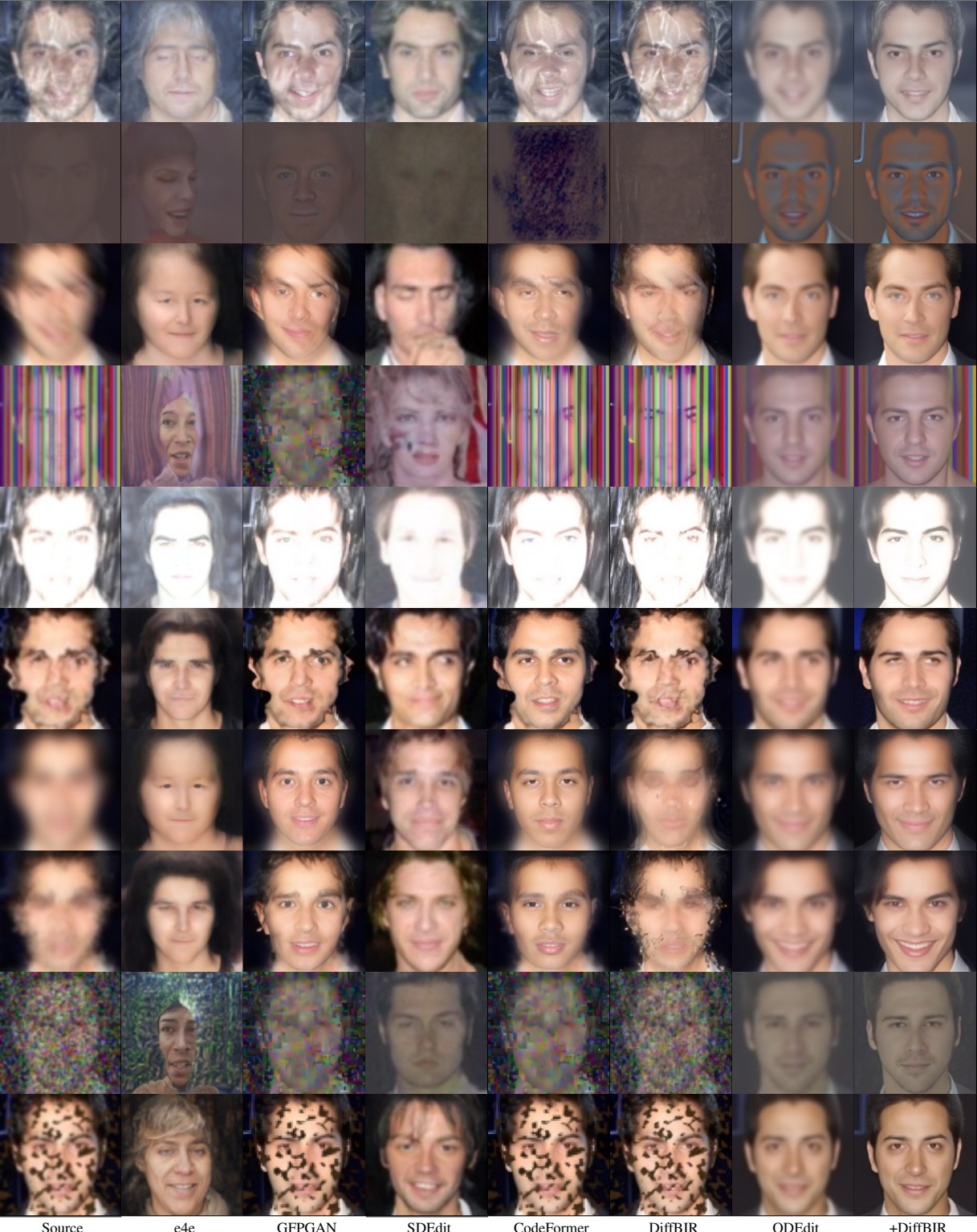

Source     e4e     GFPGAN     SDEdit     CodeFormer     DiffBIR     ODEdit     +DiffBIR

Figure 12: Visual comparison of Blind Face Restoration on severe corruptions

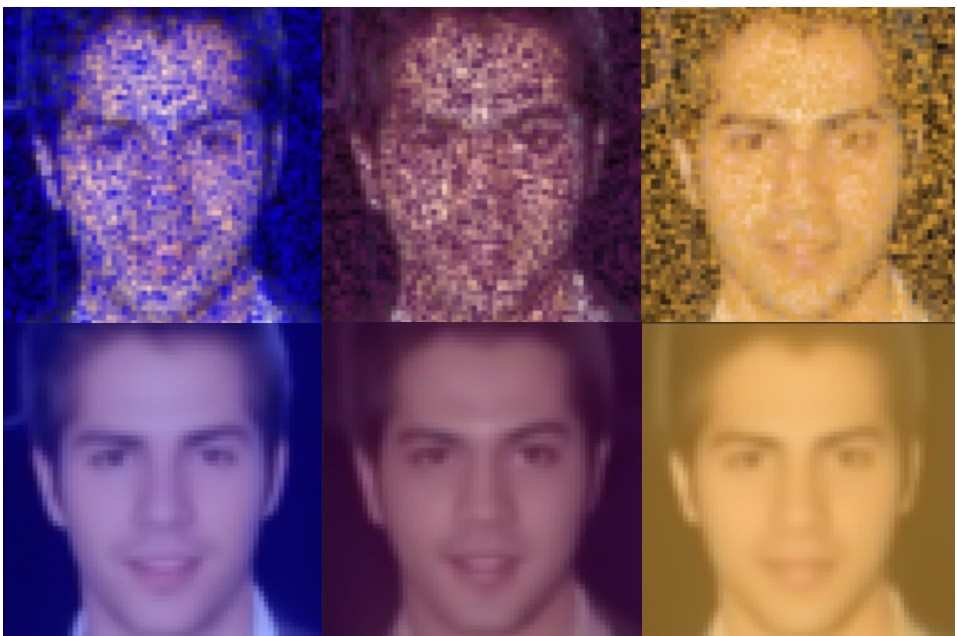

Figure 13: Examples of reconstructions showing the limits of ODEdit when facing color degradations. The top row is the observations, the bottom row is the restorations. We masked 50% of the pixels using of single color. Due to its limitations in exploring the neighborhood around the observation, ODEdit cannot recover from the color bias.

