# OpenReview forum: "ODEdit: Blind Face Restoration through Ordinary Differential Equations"
_ICLR.cc/2024/Conference — Submitted to ICLR 2024_

### Official Review · Reviewer_xfaR · 2023-10-19

**Soundness:** 2 fair
**Presentation:** 2 fair
**Contribution:** 2 fair
**Rating:** 5
**Confidence:** 4

**Summary:**

This paper proposes a method to blindly restore face images. Inspired by SDEdit, the proposed method uses ODE to circumvent the injection of extra noise. The provided experimental results show that the proposed method has some advantages over existing methods when the degradation is severe. This paper fails to describe the advantage when replacing SDE with ODE. In addition, some details about the algorithm and experiments are missing.

**Strengths:**

The provided experimental results show that the proposed method has some advantages over existing methods when the degradation is severe.

**Weaknesses:**

1. SDEdit can solve plenty of image editing and synthesis tasks. However, the proposed ODEdit is only for face restoration. Can the authors explain the reasons?
2. The advantage that replacing SDE with ODE is not clear. Why should the injection of extra noise be circumvented? Are there any disadvantages when using ODE?
3. It is not clear how can we get Algorithm 1.
4. The experiments lack some details and some comparisons are unfair and incomplete.
(1) What is the testing dataset? The authors seem not to describe it in the manuscript.
(2) Why not compare different methods on real degraded face images?
(3) The synthetic degradations are based on (Hendrycks & Dietterich, 2019) which is not considered in existing methods e.g. CodeFormer. This makes the comparison unfair. Why not compare different methods based on the testing set of CodeFormer?

**Questions:**

See weaknesses for details.

---

> ### Author Response · Authors · 2023-11-22
>
> Thank you for the review and thoughtful feedback. Below we address specific questions.
>
> Q: SDEdit can solve plenty of image editing and synthesis tasks. However, the proposed ODEdit is only for face restoration. Can the authors explain the reasons?
>
> A: The current bottleneck of ODEdit lies in the instability of DDIM inversion. The knowledge that an image represents a face acts as a strong prior, which significantly stabilizes the DDIM inversion, thereby facilitating the application of our method. However, when applied to varied-domain images such as those in ImageNet, the inversion process proved too unstable even before performing any update steps, preventing a fair assessment of the method's performance.
>
> Q: The advantage that replacing SDE with ODE is not clear. Why should the injection of extra noise be circumvented? Are there any disadvantages when using ODE?
>
> A: In fact, injecting random noise into an input increases the complexity of the denoising task, especially in extreme cases. For instance, if the input is already clean, adding noise to further clean it paradoxically introduces additional stochasticity. A major challenge with SDEdit arises when the input is so severely damaged that a significant amount of noise must be injected to generate outputs that are within the expected distribution. In such cases, fidelity to the original image may be completely lost and such scenarios are the ones we mostly address. Some papers address this issue by adding extra guidance when using SDEdit for denoising. However, we propose a guidance-free method. There are certainly drawbacks to using ODEs. Primarily, the quality of images generated using SDEs appears to be superior. Additionally, our method heavily depends on the accuracy of ODE inversion; inaccuracies in this process can lead to failure, as explained previously. Finally, our method is slower as it requires to perform multiple steps to invert the image, whereas SDEdit simply injects noise.
>
> Q: It is not clear how can we get Algorithm 1.
>
> A: We have included more details and insights about algorithm 1.
>
> Q: The experiments lack some details and some comparisons are unfair and incomplete. (1) What is the testing dataset? The authors seem not to describe it in the manuscript. (2) Why not compare different methods on real degraded face images? (3) The synthetic degradations are based on (Hendrycks & Dietterich, 2019) which is not considered in existing methods e.g. CodeFormer. This makes the comparison unfair. Why not compare different methods based on the testing set of CodeFormer?
>
> A: Our method is designed to work on 64x64 images, which limits its applicability. Since we propose a blind restoration method and are comparing it with other methods that also claim to be blind, the type of corruption should not be a concern. Therefore, it's reasonable to compare blind restoration methods across various types of corruption. However, it's important to note that some methods in our benchmark specifically incorporate super resolution and blur during training. Consequently, we expect these methods to perform better with these specific types of noise. Our goal is to use corruptions that we believe were not present during training, in order to accurately assess the real robustness of these restoration approaches. We aim to test a wide range of corruptions to evaluate the versatility and robustness of the methods, rather than their exceptional performance in specific scenarios. Additionally, we have included an example of restoration on a real, old, damaged image in the supplementary material, despite the fact that this specific sort of monochromatic damaged images is a known limit of our approach.

---

### Official Review · Reviewer_UVwz · 2023-10-28

**Soundness:** 2 fair
**Presentation:** 2 fair
**Contribution:** 2 fair
**Rating:** 3
**Confidence:** 5

**Summary:**

This paper develops an unsupervised blind face restoration method ODEdit based on a pretrained diffusion model. The ODEdit utilizes score-based updates to augment the realism of the reconstructed images. The experimental results show that ODEdit outperforms state-of-the-art blind face restoration methods in terms of fidelity and realism.

**Strengths:**

1)	The motivation of this paper is clear. The authors clearly point out that existing BFR methods still struggle to achieve faithful and realistic reconstruction.
2)	The related works is well-structured.
3)	The proposed method is the first unsupervised approach to use ODE-based inversion in Diffusion Models for Image restoration, which is innovate.

**Weaknesses:**

1)	Some parts of the paper are difficult to follow, and additional explanations and clarifications would improve the reader's understanding.
2)	The experiments are not rich and convincing. The experiments lack analysis. For example, Section 4.4 SYNERGY STUDY, the authors only present some figures without any analysis about the results.
3)	In experiments, the authors do not provide the definition of LDM. In addition, Section 4.3 and 4.4 only verify the superiority of ODEdit but cannot prove the robustness and adaptability of ODEdit as introduced in abstract.
4)	Face ﬁdelity is not only reflected in reconstruction metrics (PSNR,SSIM) but also in face recognition. Maybe the authors should provide some face recognition metrics to further verify the degree of face fidelity.

**Questions:**

1)	Compared with existing BFR methods, the proposed method is unsupervised method without any prior. Then, how does this method perform on real-world face images?
2)	The experimental part is insufficient and lack of analysis. The experimental results can only verify the superiority of the proposed method but cannot verify its robustness and adaptability. The authors should provide more convincing experiments and detailed analysis.
3)	The implementation details are missing. The authors should provide the degradation process of this paper in details.

---

> ### Author Response · Authors · 2023-11-22
>
> Thank you for the review and thoughtful feedback. Below we address specific questions.
>
> Q: Face ﬁdelity is not only reflected in reconstruction metrics (PSNR,SSIM) but also in face recognition. Maybe the authors should provide some face recognition metrics to further verify the degree of face fidelity.
>
> A: Thank you for your comment. It is indeed ideal to reconstruct a face that can be recognized as the same individual as in the source  image by face recognition algorithms. However, since our focus is on cases with a high degree of corruption, the faces in the source images are often unrecognizable. In many instances, face recognition algorithms even fail to detect a face in the source images, let alone provide meaningful landmarks. In such cases of high corruptions, comparing directly with the target is not our goal per se. We strive to generate a realistic image as close as possible to the source image.
> As a result, in cases of severely corrupted inputs, generating a realistic, plausible face already constitutes an important step.
>
>
> Q: Compared with existing BFR methods, the proposed method is unsupervised method without any prior. Then, how does this method perform on real-world face images?
>
> A: Thank you for your comment. We have included an example of an old, damaged image in our appendix. However, the prior model we trained was based on a relatively small number of faces, all of which are centered, aligned, and have pleasing colorations. Consequently, the prior we are using is somewhat limited in directly addressing real-world scenarios. Therefore, our examples should be viewed more as proofs of concept rather than as a definitive model.
>
> Q: The experimental part is insufficient and lack of analysis. The experimental results can only verify the superiority of the proposed method but cannot verify its robustness and adaptability. The authors should provide more convincing experiments and detailed analysis.
>
> A: We emphasize that our method performs more consistently across a wide range of different corruptions. This consistency is what we refer to as robustness, as opposed to excelling in handling a specific type of corruption. Indeed, when considering one type of corruption at a time, there are methods that outperform ours. However, these methods often fail more dramatically when confronted with different types of corruption. It is this versatility that we specifically aim to achieve.
>
> Q: The implementation details are missing. The authors should provide the degradation process of this paper in details.
>
> A: We have included additional details in the paper.

---

> > ### Comment · Reviewer_UVwz · 2023-11-23
> >
> > The most fundamental principle of face image restoration is that the identity information of the person cannot be changed. Regardless of the quality of the input image, we must not alter the identity of the output face. Therefore, it is important to measure the fidelity of the reconstruction results using face recognition. Additionally, I agree with Reviewer 5VT9 that the paper is about face restoration, but there is nothing about faces. The authors seem to have only applied an existing theoretical model to this problem, and their contribution to this field is limited.

---

### Official Review · Reviewer_Gq4R · 2023-10-29

**Soundness:** 4 excellent
**Presentation:** 3 good
**Contribution:** 2 fair
**Rating:** 5
**Confidence:** 4

**Summary:**

In this paper, the authors tackle the problem of blind face restoration by formulating it as an inverse problem and utilizing a diffusion model as an Ordinary Differential Equations (ODE) solver. Compared to previous methodologies, they achieved a higher Frechet Inception Distance (FID) and provided ample theoretical basis for their approach. Through various experimental results, the paper offers interpretations of the problem and the proposed solution.

**Strengths:**

- As seen in Table 1, the image quality and Frechet Inception Distance (FID) score are high.
- As observed in Figure 3, the model is robust to strong noise.
- Leveraging the generative capabilities of the diffusion model to achieve high image quality is one of the strengths of this paper.

**Weaknesses:**

- There is an identity leakage. Can this be genuinely considered "restoration" of a face? It appears to be generating a face referencing the given corrupted face image.
- As seen in Table 1, the Peak Signal-to-Noise Ratio (PSNR) and Structural Similarity Index (SSIM) are worse than previous works.

**Questions:**

If the diffusion model outputs the predicted data instead of the noise, you could potentially utilize face identity loss with a face recognition network. Would this be a feasible naive extension to address the identity leakage problem?

**Details Of Ethics Concerns:**

- The model may work for only the celebrated face. One way to figure out is utilize FairFace [1] dataset for inference.

**References**
[1] Karkkainen, K., & Joo, J. (2021). Fairface: Face attribute dataset for balanced race, gender, and age for bias measurement and mitigation. In Proceedings of the IEEE/CVF winter conference on applications of computer vision (pp. 1548-1558).

---

> ### Author Response · Authors · 2023-11-22
>
> Thank you for the review and thoughtful feedback. Below we address specific questions.
>
> Q: If the diffusion model outputs the predicted data instead of the noise, you could potentially utilize face identity loss with a face recognition network. Would this be a feasible naive extension to address the identity leakage problem?
>
> A: Thank you very much indeed for your suggestion. Introducing guidance into the generative process of ODEdit is certainly feasible, and employing a face recognition network with a face identity loss like cosine similarity, could effectively address the identity leakage issue. This approach has been successfully implemented in a recent paper. However, our aim is to design a method robust enough for extreme scenarios where other methods might fail. For guidance to be effective in such cases, the face recognition network itself must be equally robust. Generally, adding any guidance necessitates the guidance method to be as resilient as our method to add value. Practically, in low-noise conditions, this works quite well, but in such scenarios, our method does not outperform others anyway. Conversely, in high-noise situations, we observed that face detection might fail, and landmarks can be unreliable. Considering these extreme cases, we currently prefer a method that relies solely on itself. Furthermore, while identity leakage is a significant concern generally, in cases of severely corrupted inputs, being able to generate a realistic, plausible face is an important initial step.
>
> Q: The model may work for only the celebrated face. One way to figure out is utilize FairFace [1] dataset for inference.
>
> A: Thank you for your comment. It is absolutely right; however, our paper primarily focuses on using a pretrained model to restore images, rather than on the training procedure itself. As a result, if the pretrained model contains biases, they will indeed be reflected in our method. However, we do not believe this fundamentally challenges our method per se. Replacing the pretrained model with an unbiased one would be a straightforward solution.
>
> Q: As seen in Table 1, the Peak Signal-to-Noise Ratio (PSNR) and Structural Similarity Index (SSIM) are worse than previous works.
>
> A: Our method aims to make as few changes as possible to the source image in order to enhance its realism. Consequently, all pixel-based metrics remain close to those of the corrupted image, while the FID and LPIPS metrics show significant improvement. This is a direct result of our approach.

---

### Official Review · Reviewer_5VT9 · 2023-10-30

**Soundness:** 2 fair
**Presentation:** 2 fair
**Contribution:** 2 fair
**Rating:** 3
**Confidence:** 3

**Summary:**

This paper introduces a method for blind face restoration using a pretrained diffusion model. The main idea is to formulate the inverse problem using an ODE for image inversion. The method is evaluated on several image restoration tasks involving different amounts of corruptions and noise levels.

**Strengths:**

* The method is simple.
* The method produces results that are better than the compared methods when plugged with another prior.

**Weaknesses:**

* Writing and presentation needs to be significantly improved before we can judge the real contributions.
* The motivation for the method is unclear and hand-wavy.
* Results are not clear: the base method leads to blurry reconstructions.
* The paper is about face restoration, but there's nothing about faces -the only thing about faces is that the base diffusion model is trained on faces.

**Questions:**

The paper introduces a method for sampling a new image from a diffusion prior that is compatible to an observation. This is done through first estimating a latent code of the distorted image and then resampling a new sample starting from that latent code. The method is simple. The main issue is that the presentation is cumbersome.

I have the following questions/comments:
1.  **Algorithm 1**. This is the core of the method. It's unclear what does it mean  ODE_SOLVER( . ). Is this running one step of DDIM Eq(9)? This is used twice in the algorithm. The rationale of the algorithm is not clear or explained.

2. **Results of the base methods are blurry**. So this method doesn't seem to be doing a good job. Better results are obtained when plugging the method with another prior (CodeFormer, GFPGAN, etc). So in the end, it's more like with this ODEdit you can get a blurry reconstruction (Average) and then from there another prior is enforced to get a better reconstruction. So in the end, where is the gain coming from? Also it's not clear how this method is applied in conjunction with other methods.

3. **Presentation is cumbersome**. The method is presented in page 6. The first 5 pages are related to some background. However, a big part of the background is irrelevant and other relevant background (like a clean explanation of SDEdit) is not given. This leads to an unbalanced presentation that is hard to follow, and hard to fairly judge the real contributions of the paper.

---

> ### Author Response · Authors · 2023-11-22
>
> Thank you for the review and thoughtful feedback. Below we address specific questions.
>
> Q: Algorithm 1. This is the core of the method. It's unclear what does it mean ODE_SOLVER( . ). Is this running one step of DDIM Eq(9)? This is used twice in the algorithm. The rationale of the algorithm is not clear or explained.
>
> A: We clarified in the paper what is meant by ODE_solver. In our practical case, it consists indeed of running one step of DDIM. It is used twice because it is used backward and forward, to revert the image, and generate a new one, as an ODE can be run both ways. We added more explanations regarding algorithm 1.
>
>
> Q: Results of the base methods are blurry. So this method doesn't seem to be doing a good job. Better results are obtained when plugging the method with another prior (CodeFormer, GFPGAN, etc). So in the end, it's more like with this ODEdit you can get a blurry reconstruction (Average) and then from there another prior is enforced to get a better reconstruction. So in the end, where is the gain coming from? Also it's not clear how this method is applied in conjunction with other methods.
>
> A: Our method works with 64x64 images, whereas other methods perform super resolution in 512x512. As a result, ODEdit outputs do seem low resolution compared to others because they are. However, ODEdit is not simply producing blurry or averaged reconstruction. We proposed adding a super resolution model post-ODEdit to compare improvements more fairly. Furthermore, other models simply fail to work on severe corruption altogether, while ODEdit makes them work again. We chose to focus on 64x64 images because the ODE solver we used (DDIM) tends to be more and more unstable when moving to higher resolution. However, provided that DDIM inversion works, we added some results we obtained in 256x256 in Appendix that show a significantly higher degree of details. We also added in Appendix example comparing plugging blurred images, as suggested, to plugging ODEdit outputs to super resolution models. Finally, to combine the methods, we only need to input low-resolution ODEdit outputs to other approaches.
>
> Q: Presentation is cumbersome.
>
> A: We have made modifications in order to follow your comments, reducing the background to more closely related research and extending our description of SDEdit.
>
> Q:The paper is about face restoration, but there's nothing about faces -the only thing about faces is that the base diffusion model is trained on faces.
>
> A: Thank you for your comment. It is absolutely right, nothing seems to prevent us from transferring the proposed approach to other domains apart from the instability of the DDIM inversion. When we applied it to other domains such as Imagenet, even reconstructions on clean data could be inaccurate, hence our choice to first narrow down our focus on faces.

---

> > ### Comment · Reviewer_5VT9 · 2023-12-04
> > **Re: Official Comment by Authors**
> >
> > I appreciate the authors' efforts in addressing my previous concerns. However, I believe the major points I raised in my review remain unaddressed, which leads me to maintain my initial score. While the paper shows promise, it still feels premature in its current form. Additional evidence and a clearer presentation would be necessary to fully assess its potential contributions.

---

### Meta-Review · Area_Chair_BA79 · 2023-12-13

**Metareview:**

The paper proposes to use Ordinary Differential Equations (ODE) for the blind face image restoration task. The experimental results show that the method has some benefits when the input has severe degradations.
Strengths: (1) propose to replace SDE with ODE, (2) the method seems to be effective for input with severe degradations.
Weaknesses: (1) the paper writing needs improvement! (2) the advantage of replacing SDE with ODE is not clear and convincing, (3) incomplete and unfair experiments.
The paper should be improved by considering the weaknesses mentioned above.

**Justification For Why Not Higher Score:**

The paper got all negative feedbacks from four expert reviewers.

**Justification For Why Not Lower Score:**

N/A

---

### Decision · Program_Chairs · 2024-01-16

Reject